# A High Methylation Level of a Novel −284 bp CpG Island in the *RAMP1* Gene Promoter Is Potentially Associated with Migraine in Women

**DOI:** 10.3390/brainsci12050526

**Published:** 2022-04-21

**Authors:** Estefânia Carvalho, Andreia Dias, Alda Sousa, Alexandra M. Lopes, Sandra Martins, Nádia Pinto, Carolina Lemos, Miguel Alves-Ferreira

**Affiliations:** 1Instituto de Investigação e Inovação em Saúde (i3S), Universidade do Porto, 4200-135 Porto, Portugal; ecarvalho@i3s.up.pt (E.C.); andreia.dias@ibmc.up.pt (A.D.); absousa@icbas.up.pt (A.S.); alopes@ipatimup.pt (A.M.L.); smartins@ipatimup.pt (S.M.); npinto@ipatimup.pt (N.P.); clclemos@ibmc.up.pt (C.L.); 2Institute of Molecular Pathology and Immunology of the University of Porto (IPATIMUP), 4200-135 Porto, Portugal; 3Instituto Ciências Biomédicas Abel Salazar (ICBAS), Universidade do Porto, 4050-313 Porto, Portugal; 4Unit for Genetic and Epidemiological Research in Neurological Diseases (UnIGENe), Instituto de Biologia Molecular e Celular (IBMC), Universidade do Porto, 4200-135 Porto, Portugal; 5Centre for Predictive and Preventive Genetics (CGPP), Instituto de Biologia Molecular e Celular (IBMC), Universidade do Porto, 4200-135 Porto, Portugal; 6Centro de Matemática da Universidade do Porto (CMUP), 4169-007 Porto, Portugal

**Keywords:** DNA methylation, epigenetics, migraine, *RAMP1*

## Abstract

Migraine is a complex neurovascular disorder affecting one billion people worldwide, mainly females. It is characterized by attacks of moderate to severe headache pain, with associated symptoms. Receptor activity modifying protein (RAMP1) is part of the Calcitonin Gene-Related Peptide (CGRP) receptor, a pharmacological target for migraine. Epigenetic processes, such as DNA methylation, play a role in clinical presentation of various diseases. DNA methylation occurs mostly in the gene promoter and can control gene expression. We investigated the methylation state of the *RAMP1* promoter in 104 female blood DNA samples: 54 migraineurs and 50 controls. We treated DNA with sodium bisulfite and performed PCR, Sanger Sequencing, and Epigenetic Sequencing Methylation (ESME) software analysis. We identified 51 CpG dinucleotides, and 5 showed methylation variability. Migraineurs had a higher number of individuals with all five CpG methylated when compared to controls (26% vs. 16%), although non-significant (*p* = 0.216). We also found that CpG −284 bp, related to the transcription start site (TSS), showed higher methylation levels in cases (*p* = 0.011). This CpG may potentially play a role in migraine, affecting *RAMP1* transcription or receptor malfunctioning and/or altered CGRP binding. We hope to confirm this finding in a larger cohort and establish an epigenetic biomarker to predict female migraine risk.

## 1. Introduction

Migraine is a common debilitating neurological disease characterized by severe throbbing pain lasting from 4 to 72 h, usually accompanied by nausea, vomiting, photophobia, and phonophobia; in migraine of aura subtype, an aura may precede or occur during the attack [1].

According to the Global Burden of Disease study published in 2017, migraine affects nearly 1.25 billion (10^9^) people worldwide and was ranked the second most disabling disease in terms of years lived with disability [2]. Importantly, women are significantly more affected by migraine than men [3].

Migraine is a complex disease due to its multifactorial inheritance since; in addition to being polygenic, it also has an environmental component, contributing to the difficulty in understanding the underlying pathophysiologic processes [4]. 

It is generally believed that a key component of migraine pathophysiology is the activation and sensitization of the trigeminovascular system, which leads to the release of neuropeptides, mainly the Calcitonin gene-related peptide (CGRP), triggering neurogenic inflammation [5]. This hypothesis is supported by the detection of elevated CGRP blood levels during active migraine episodes, as well as in chronic versus episodic migraineurs [6,7]. The importance of CGRP in migraine is also supported by the success of CGRP antagonists or antibodies against CGRP in the treatment or prevention of the disease [8,9,10,11,12,13].

CGRP is a potent vasodilator neuropeptide, synthesized in neurons through alternative splicing of mRNA transcribed from the *CALCA* gene (11p15.2), and is widely produced throughout the central and peripheral nervous systems [14,15]. 

CGRP mediates its effects through a receptor that consists of the calcitonin receptor-like receptor (CLR) and the receptor activity-modifying protein (RAMP1). A third constituent, the receptor component protein (RCP), is an accessory protein necessary for proper function since it connects the receptor to downstream signaling pathways [16,17].

Since RAMP1 plays a pivotal role in the CGRP receptor, one can speculate that alterations in RAMP1 expression might influence CGRP receptor expression or function. In fact, it has been reported that altered RAMP1 levels prompt changes in CGRP receptor activity [18,19]. 

The activity of a given gene may be altered through epigenetic changes, as a result of modifications in the way the DNA sequence is read (not involving alterations in the nucleotide sequence). While these changes are natural and essential for normal development, some may also have adverse effects and cause abnormal gene activation or silencing [20]. Previously, epigenetic processes have been shown to play a role in various diseases, including neurological diseases such as migraine [21,22,23]. Indeed, aberrant epigenetic patterns can be used as biomarkers for the diagnosis and prognosis of many diseases, and may even be capable of distinguishing different subtypes of a specific disease [24,25,26,27,28].

Possibly the most well-known epigenetic process is DNA methylation. Briefly, DNA methylation occurs mostly at cytosine residues in CpG dinucleotides in the gene promoter and can regulate gene expression either by recruiting proteins involved in gene repression or hindering the binding of transcription factors to DNA [29]. 

Our aim in this study was to investigate the *RAMP1* promoter methylation status in female migraineurs and controls in order to find epigenetic biomarkers that can predict migraine risk in an accessible body fluid, such as blood.

## 2. Materials and Methods

### 2.1. Subjects and Study Design

This case-control study was conducted in a female cohort selected at the outpatient neurologic clinic at Centro Hospitalar Universitário do Porto (CHUP) and at Centre for Predictive and Preventive Genetics (CGPP). A total of 104 samples were analyzed—54 samples of women diagnosed with migraine and 50 migraine-free women control samples. At the time of observation, the mean age of controls was higher than that of migraineurs (36.2 ± 12.4 vs. 32.3 ± 11.4 years), which gives us confidence that controls are, in fact, migraine-free (Table 1). Patients with familial hemiplegic migraine were excluded; those with the co-occurrence of migraine with aura (MA) and migraine without aura (MO) were included in the MA group, as done in other studies [30,31]. Controls and cases were from the same ethnic and geographical origin, age-matched, and non-related.

All cases and controls underwent a diagnostic interview, using the same structured questionnaire, based on the operational criteria of the International Headache Society (IHS)—3rd edition of the International Classification of Headache Disorders (ICHD-3) [1,32]. Blood samples were collected in the sequence of a neurologic appointment and were kept at CGPP’s biobank. Additional clinical information of subjects was collected, allowing the exclusion of those with potential confounding diagnoses. Women with menstrual headaches were excluded from the control group. 

The Ethics Committee of CHUP approved the study and participants were asked to give their written informed consent to enroll in the study. 

### 2.2. DNA Extraction and Bisulfite Conversion

DNA extraction from blood samples was performed by the standard salting-out method using QIAamp^®^ DNA Blood Mini Kit [33]. DNA quantification was performed using Nanodrop 2000.

DNA samples were converted using the EZ DNA Methylation-Lightning Kit (Zymo Research, Irvine, CA, USA), according to the manufacturer’s instructions. 

### 2.3. PCR and Sanger Sequencing

After bisulfite conversion, PCR amplification of the promoter region was performed using a pair of primers that allowed the analysis of 399 and 94 bases upstream and downstream of the transcription start site (TSS), respectively (ENST00000254661.5 transcript). Sequences are available upon request.

Amplified products were sequenced through Sanger Sequencing, by using the Big Dye^®^ Terminator Cycle Sequencing 1.1 Ready Reaction kit (Applied Biosystems, Waltham, MA, USA) and the same primers used in the PCR, following the manufacturer’s instructions; products were then loaded in an ABI-PRISM 3130 XL Genetic Analyzer (AppliedBiosystems, Waltham, MA, USA).

### 2.4. Methylation Analysis

Sequences were analyzed for CpG island methylation using the Epigenetic Sequencing Methylation Analysis (ESME) software from Epigenomics AG. The ESME software performs quality control, corrects incomplete conversion, normalizes signals, and provides the measurement of cytosine methylation by comparing the C and T peaks at CpG sites [34]. The calculated methylation calls by the ESME software were reviewed and inspected by using the associated electropherograms generated by the software. 

To determine a methylation signal threshold, we converted and sequenced purified methylated and non-methylated human DNA sets that we used as positive and negative controls, respectively (Human WGA Methylated & Non-methylated DNA Set, Zymo Research). We analyzed the methylated control (enzymatically methylated at all double-stranded CG dinucleotides) and achieved results of nearly 100% methylation levels, as expected. When analyzing the non-methylated control (purified from cells that contain genetic knockouts of both DNA methyltransferases DNMT1 (-/-) and DNMT3b (-/-)), we still observed methylation levels as high as 19%. Thus, we considered these as sequencing or ESME artifacts and determined a 20% threshold. As such, signals under 20% were considered non-methylated and signals higher than 20% were considered methylated. This cut-off value was also established in other studies [35,36]. This means that all subsequently analyzed CpG units were divided into “methylated” or “non-methylated” based on the methylation signals measured by ESME software.

### 2.5. Statistical Analysis

To compare the frequencies of methylated/non-methylated status of the five studied CpG units in cases and controls (with the cut-off described above), we performed a chi-squared test. Power analyses previously developed using R software, pwr package [37,38], showed that our sample has an associated power greater than 85% for medium and large effect sizes (>0.3) and a significance threshold equal to 0.05 (Table 2). The power of our sample (N = 104) to detect a small effect size (0.1) is equal to 18%.

We also performed a logistic regression where status (cases vs. controls) was the dependent variable and the methylation levels for each of the five CpG units were the independent variable (adjusting for age at observation) to estimate the odds ratio and the respective confidence intervals. The significance threshold was set at α = 0.05; statistical analyses were performed using SPSS version 27.

## 3. Results

### 3.1. Study Demographics

Demographic data from patients and controls are shown in Table 1. All subjects were female and the case-control ratio was 1.1:1. Given our limited sample size, we did not stratify migraineurs into subgroups, such as MA or MO. 

### 3.2. Promoter Methylation Profile

A total sequence of 493 bp was amplified, containing the *RAMP1* core promotor and the beginning of exon 1. We identified 51 CpG dinucleotides, but only the first 5 showed different methylation status in our samples (−346, −334, −284, −276, −234, related to the TSS), with the other 46 units being consistently unmethylated (Figure 1).

### 3.3. CpG Unit Methylation

We found a higher proportion of migraine cases with all five CpG units methylated (26% of cases had the five units methylated in contrast with only 16% of controls). However, this proportion did not differ significantly between cases and controls (*p* = 0.216).

Afterwards, we assessed the methylation levels of DNA chains for the CpG units (Table 3). The −284 CpG unit showed significantly higher methylation levels in cases when compared to controls (*p* = 0.011, OR = 1.07; 95% C.I.: 1.02–1.12), which supports the hypothesis that migraineurs have more DNA chains methylated at unit −284 than controls (Table 3).

## 4. Discussion

To properly exert its function, CGRP requires a receptor CLR, a subunit designated RAMP1, and a small intracellular protein RCP. RAMP1 is essential for CGRP activity as it is required for trafficking CLR to the cell surface and confers CGRP binding specificity [16]. A relevant role for RAMP1 in migraine pathology has been shown in several studies. In 2007, Zhang and colleagues generated mice expressing human RAMP1 in the nervous system and demonstrated that RAMP1 levels are functionally rate limiting in the trigeminal ganglia and that elevation of RAMP1 increases neuronal CGRP receptor activity and CGRP-induced subcutaneous inflammation [19].

It has also been reported that CGRP-sensitized mice overexpressing RAMP1 show aversion to light, similarly to photophobia, which is a well-known migraine symptom [40,41].

We identified a novel CpG unit in the *RAMP1* promoter (−284 related to the TSS), significantly methylated in migraineurs. We found no significant differences between global promoter methylation of cases and controls, although migraineurs had a higher proportion of individuals with all five CpG units methylated. At this point it should be noted that, despite appropriate-to-detect medium-to-large size effects, power analyses showed that nearly eightfold our sample size would be needed to detect small effects with a power of 80% (Table 2). 

To the best of our knowledge, only one study has previously analyzed the methylation status of the human *RAMP1* promoter [42]. Wan et al. studied the *RAMP1* promoter in blood samples of 26 migraineurs and 25 controls, including males and females. Although the authors did not find statistically significant differences between cases and controls for the methylation levels of any of the 13 CpG sites analyzed (α = 0.05, *p* = 0.197), they found a low methylation trend in migraine cases, as well as higher and lower methylation levels associated with positive migraine family history (+25, +27, +31, related to the TSS) and female migraineur versus healthy female (+89, +94, +96), respectively. 

Interestingly, our results seem to contradict this study [42], as we observed higher promoter methylation levels in female migraineurs when compared to female controls. Unfortunately, we were unable to compare the sequence analyzed by Wan and peers; therefore, we were incapable of precisely matching our results with those obtained in their study. While they studied a sequence ranging from −300 to +205, related to the TSS, and identified 13 CpG units, our sequence ranged from −399 to +94 and identified 51 CpG units. Since we were unable to identify the same CpG units, we assume that the TSS they identified is located in a different region.

Here, we describe a CpG unit (−284 in relation to the TSS) that shows significantly higher methylation levels in migraineurs and may thus play a role in migraine susceptibility. 

DNA methylation has been implicated in several diseases and it is well known that changes in methylation patterns may affect gene expression [22]. In fact, Park and collaborators set out to investigate whether *CALCA* gene expression was influenced by epigenetic alterations using rat and human cell lines, and rat trigeminal glial cultures [43]. They measured DNA methylation, as well as histone acetylation, at CpG islands located in the promoter region. The authors found that DNA methylation correlates with *CALCA* gene expression, as the CpG islands analyzed were hypermethylated in cells not expressing the gene and hypomethylated in cells expressing *CALCA* [43]. It is important to note that although DNA methylation is typically associated with gene repression, some studies have revealed that it can actually lead to upregulation of gene expression [44,45]. 

Thus, although it is tempting to suppose that the hypermethylation of *RAMP1* may hinder or promote its transcription or alter its proper function, further studies are needed to address this issue.

The present findings represent a first step towards the establishment of an epigenetic biomarker that can predict migraine risk in women by simple DNA analysis in an accessible body fluid.

To strengthen and validate our findings, further studies should be performed in larger cohorts to investigate *RAMP1* expression and biochemistry to determine whether the −284 CpG unit methylation affects *RAMP1* transcription or whether it leads to receptor malfunctioning and/or altered CGRP binding.

## 5. Conclusions

As CGRP is a target for migraine treatment, understanding the CGRP pathway is an important step to understanding migraine. The CGRP receptor includes three proteins, one of which is RAMP1.

To date, only one other study has analyzed the methylation of the human *RAMP1* gene promoter in the context of migraine. In this study, we found a novel CpG unit significantly associated with the disease. This CpG could potentially serve as a biomarker capable of identifying migraine susceptibility in females. However, given the size limitation of our cohort, further studies including more samples will need to be done to confirm and strengthen our findings.

## Figures and Tables

**Figure 1 brainsci-12-00526-f001:**
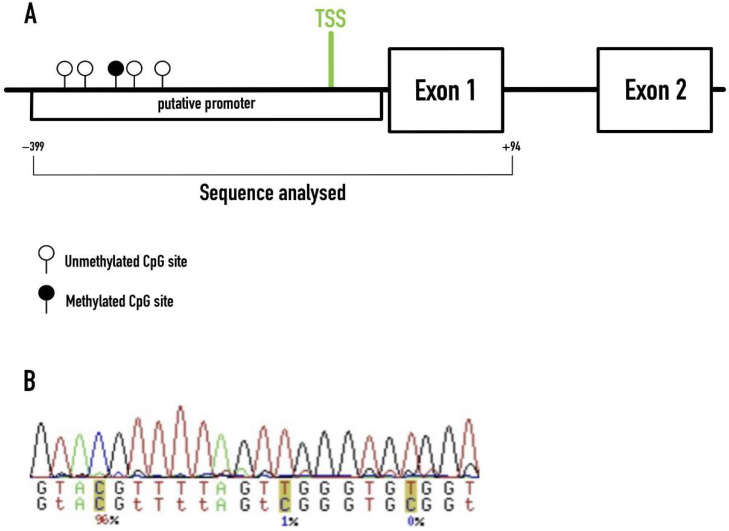
Methylation profiling of the *RAMP1* promoter region in our cohort. (**A**) Sequence of the *RAMP1* gene analyzed in this study. The methylated CpG site illustrated (black circle) refers to the −284 CpG unit found to be significantly methylated in migraineurs. TSS–Transcription Start Site. (**B**) Example of the ESME software output.

**Table 1 brainsci-12-00526-t001:** Demographic and clinical data of migraine patients and controls.

	Migraine Patients	Controls
Females (n)	54	50
MA	28 (51.9%)	n/a
MO	26 (48.1%)	n/a
Mean age at observation (±SD) –years	32.3 (±11.4)	36.2 (±12.4)

SD—standard deviation; MA—migraine with aura; MO—migraine without aura; n/a—not applicable

**Table 2 brainsci-12-00526-t002:** Sample size requirements for the chi-squared test for association for different effect sizes and powers, in case-control studies (df = 1) for a level of significance equal to 0.05.

Effect Sizes *	Power			
	80%	85%	90%	95%
Small (0.1)	785	898	1051	1300
Medium (0.3)	88	100	117	145
Large (0.5)	32	36	43	52

* Conventional effect sizes as established by Cohen for the chi-squared test [39].

**Table 3 brainsci-12-00526-t003:** Results from a multivariable logistic regression.

	OR	95% C.I.	*p*-Value
CpG −346	0.99	0395–1.03	0.598
CpG −334	1.01	0.96–1.06	0.760
CpG −284	1.07	1.02–1.12	0.011 *
CpG −276	0.97	0.92–1.02	0.234
CpG −234	0.98	0.92–1.03	0.410
Age at observation	0.98	0.95–1.01	0.203

OR—odds ratio; C.I.—confidence interval; * *p* < 0.05.

## Data Availability

Not applicable.

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
