# Peer review of "A High Methylation Level of a Novel −284 bp CpG Island in the RAMP1 Gene Promoter Is Potentially Associated with Migraine in Women"

_brainsci, 2022, doi:10.3390/brainsci12050526_

Round 1
Reviewer 1 Report
This is a well written manuscript investigating methylation in the RAMP1 promoter region in association to migraine. My only concern is the low nomber of study participants. Is it possible to include more subjects in the study? If not the authors should perform a power analysis and carefully discuss the outcome of this in the manuscript. Please also add a section in the discussion where the results are discussed with the respect of the outcome of the power analysis, and the disadvantage of analyzing so few individuals.
Author Response
This reviewer is positive about the manuscript. The concerns about the low number of study participants are completely understandable. Since we were unable to include more participants during the time frame of our project, we followed the reviewer’s suggestion and presented the power analysis previously developed for our study (Material and Methods section; Statistical Analyses subsection). Some of these results are now also included in the Discussion section. In the Conclusions section, we now acknowledge that further studies including more samples should be developed to strengthen our findings.
Reviewer 2 Report
In this study, the authors conducted a case-control study to investigate the role of methylation state of the 20 RAMP1 promoter in migraine risk. This is a well written paper. I have several queries and comments in relation to the results as presented:
Major:
- When conducting logistic regression (line 132-135), the authors did not adjust age as a covariate. As aging is strongly correlated with changes in DNA methylation, some results are possibly biased. The authors should check the correlations between average methylation level and age.
- The authors are lax in correcting for false discovery, their highlight (-284 CpG unit) seems to be non-significant after multiple test corrections (line 157). The statements in the title and abstract should be, accordingly, downplayed a bit.
- The authors should add more details in the section of blood sample collection and DNA isolation and extraction.
Minor:
- Line 15: change 109 to 1 billion.
- Line 25: this comparison result is actually non-significant.
- Line 91: typo “ICDH”.
- The authors should explain more about why choosing a 20% methylation cut-off (line 123). Although it has been used in other studies (ref #35, #36), I wonder what is the results based on other cut-offs (e.g. 10%, 30%, etc).
- Describe what is alpha mean (line 182).
Author Response
Again, this reviewer is generally positive about the manuscript. We will address each of the comments individually.
Major Revisions
- When conducting logistic regression (line 132-135), the authors did not adjust age as a covariate. As aging is strongly correlated with changes in DNA methylation, some results are possibly biased. The authors should check the correlations between average methylation level and age.
We appreciate this comment. Aging is in fact closely tied to DNA methylation and so we adjusted our logistic regression analysis to age at observation to check for an association between age and methylation levels, but results were not statistically significant (p=0.203).
Given this analysis, we have replaced Table 3 to better present our results.
- The authors are lax in correcting for false discovery, their highlight (-284 CpG unit) seems to be non-significant after multiple test corrections (line 157). The statements in the title and abstract should be, accordingly, downplayed a bit.
Multiple testing correction can be extremely conservative and sometimes it is far more likely to lead to Type II errors than Type I. We are therefore confident about our results, but we reinforce in the discussion and conclusions that they need to be confirmed by other studies.
Furthermore, all CpG units were analysed in the same and unique logistic regression model and not independently, and so we set the level of significance at 0.05, since the CpG units were not analysed one by one.
The title and abstract were altered to downplay our findings, as suggested by the reviewer.
- The authors should add more details in the section of blood sample collection and DNA isolation and extraction.
As requested, we have added more information regarding this section. We hope is helps to better understand the procedures.
Minor Revisions
- Line 15: change 109 to 1 billion.
As requested, we have performed this alteration.
- Line 25: this comparison result is actually non-significant.
We appreciate the reviewer’s comments and we modified the abstract accordingly.
- Line 91: typo “ICDH”.
We thank the reviewer for this observation. We have now corrected the text accordingly.
- The authors should explain more about why choosing a 20% methylation cut-off (line 123). Although it has been used in other studies (ref #35, #36), I wonder what is the results based on other cut-offs (e.g. 10%, 30%, etc).
We appreciate the reviewer’s comment and we hope that our explanation elucidates this issue.
- Describe what is alpha mean (line 182).
We acknowledge the comment. Alpha is the level of significance, a threshold value used to judge whether a test statistic is statistically significant. We have proceeded to alter the text in a way that hopefully clarifies this matter.
Round 2
Reviewer 2 Report
The authors have addressed all my concerns.